

# The effect of captivity on the skin microbial symbionts in three *Atelopus* species from the lowlands of Colombia and Ecuador

Sandra V. Flechas[1], Ailin Blasco-Zúñiga[2], Andrés Merino-Viteri[3], Valeria Ramírez-Castañeda[1], Miryan Rivera[2] and Adolfo Amézquita[1]

[1] Department of Biological Sciences, Universidad de los Andes, Bogotá, Colombia
[2] Laboratorio de Investigación en Citogenética y Biomoléculas de Anfibios (LICBA), Centro de Investigación para la Salud en América Latina (CISeAL), Pontificia Universidad Católica del Ecuador, Quito, Ecuador
[3] Museo de Zoología (QCAZ), Escuela de Ciencias Biológicas, Pontificia Universidad Católica del Ecuador, Quito, Ecuador

Corresponding author
Sandra V. Flechas,
vickyflechas@gmail.com

## ABSTRACT

Many amphibian species are at risk of extinction in their natural habitats due to the presence of the fungal pathogen *Batrachochytrium dendrobatidis* (*Bd*). For the most highly endangered species, captive assurance colonies have been established as an emergency measure to avoid extinction. Experimental research has suggested that symbiotic microorganisms in the skin of amphibians play a key role against *Bd*. While previous studies have addressed the effects of captivity on the cutaneous bacterial community, it remains poorly studied whether and how captive conditions affect the proportion of beneficial bacteria or their anti-*Bd* performance on amphibian hosts. In this study we sampled three amphibian species of the highly threatened genus, *Atelopus*, that remain in the wild but are also part of *ex situ* breeding programs in Colombia and Ecuador. Our goals were to (1) estimate the diversity of culturable bacterial assemblages in these three species of *Atelopus*, (2) describe the effect of captivity on the composition of skin microbiota, and (3) examine how captivity affects the bacterial ability to inhibit *Bd* growth. Using challenge assays we tested each bacterial isolate against *Bd*, and through sequencing of the 16S rRNA gene, we identified species from thirteen genera of bacteria that inhibited *Bd* growth. Surprisingly, we did not detect a reduction in skin bacteria diversity in captive frogs. Moreover, we found that frogs in captivity still harbor bacteria with anti-*Bd* activity. Although the scope of our study is limited to a few species and to the culturable portion of the bacterial community, our results indicate that captive programs do not necessarily change bacterial communities of the toad skins in a way that impedes the control of *Bd* in case of an eventual reintroduction.

# INTRODUCTION

Global amphibian declines have been attributed to a variety of factors including habitat destruction, contamination, UV-B radiation, climate change, overexploitation, and

infectious diseases (*Alford & Richards, 1999*; *Collins & Crump, 2009*; *Heard et al., 2013*). Diseases caused by microparasites have been linked to massive mortality events and extinctions, and the causative agents can be viral, bacterial, or caused by protists and fungi (*Latney & Klaphake, 2013*; *Chambouvet et al., 2015*). Emerging diseases caused by fungi have become more problematic and the impact on the affected groups are much more alarming (*Wake & Vredenburg, 2008*; *Fisher et al., 2012*). *Batrachochytrium dendrobatidis* (*Bd*) (*Longcore, Pessier & Nichols, 1999*), and *B. salamandrivorans* (*Bsal*) (*Martel et al., 2013*) are highly pathogenic fungi, recognized as the etiological agents of the amphibian infectious disease known as chytridiomycosis. *Bsal* has been detected only in urodels (salamanders and newts) (*Martel et al., 2013*; *Martel et al., 2014*), while *Bd* is considered a generalist pathogen found in a wide variety of amphibian species across the three orders (i.e., Anura, Urodela and Gymnophiona). *Bd* has been detected in at least 520 amphibian species in 56 countries (*Berger et al., 2016*) causing population declines in at least 200 species across five continents (*Skerratt et al., 2007*; *Fisher, Garner & Walker, 2009*).

Conservation strategies focused on ameliorating the lethal effects of the fungal pathogen have been mainly restricted to establishing disease-free *ex situ* colonies, with the aim of ensuring the survival of susceptible species (*Becker et al., 2014*; *Tapley et al., 2015*). Captive programs are a short-term intervention to prevent extinction where the final goal is to reintroduce the animals to their native habitat (*Mendelson et al., 2006*). Although, a strategy that ensures host survival despite *Bd* presence is still needed. Meanwhile, *ex situ* programs may be the best option for many threatened amphibians occurring in complex environments, where *in situ* strategies are difficult to apply (*Bosch et al., 2015*). The conditions under which amphibians are maintained in captivity may disturb their associated symbiotic microbes (*Kueneman et al., 2016*; *Loudon et al., 2014*), which would affect health status (*Li et al., 2008*; *Becker et al., 2015*) and survival rates if released into the wild (*Redford et al., 2012*; *Michaels, Downie & Campbell-Palmer, 2014*).

Cutaneous microbes can inhibit or delay the growth of *Bd* (*Harris et al., 2006*; *Harris et al., 2009*; *Woodhams et al., 2007*), and therapies where beneficial microbes are augmented in the host skin have been considered as an option to treat *Bd*-susceptible species *in situ* (*Woodhams et al., 2011*; *Bletz et al., 2013*; *Woodhams et al., 2016*). However, new approaches should take into account the complex interactions among the host, its symbiotic bacteria and the fungal pathogen without neglecting the role of the environmental context on these interactions (*McKenzie et al., 2012*; *Kueneman et al., 2014*; *Bosch et al., 2015*). Captivity imposes a different environment that might affect the microbial composition, and thus, the response of the host to the pathogen. Various studies have suggested that skin associated microbes of individuals in captivity differ from those in the wild (*Becker et al., 2014*; *Kueneman et al., 2016*; *Sabino-Pinto et al., 2016*). In the case of the red-eyed tree frog (*Agalychnis callidryas*), community composition, species richness, and abundance of bacterial groups seem to be influenced mainly by the availability of carotenoids in their diet (*Antwis et al., 2014*), and by the cover provided in the enclosures (*Michaels, Antwis & Preziosi, 2014*). However, other factors including humidity, temperature, pH and disinfection methods affect the presence of certain bacterial species and could facilitate or impede colonization and establishment in the host skin (*Mendoza et al., 2012*). Some studies

have demonstrated that OTU (Operational Taxonomic Units) richness and phylogenetic diversity are significantly higher in captive animals (i.e., in *Atelopus zeteki, Becker et al., 2014*), while other studies find the opposite, where wild animals harbor a higher bacterial diversity (i.e., *Cynops pyrrhogaster, Sabino-Pinto et al., 2016* and, *Anaxyrus boreas, Kueneman et al., 2016*).

Although changes in the bacterial assemblage composition due to captivity seem to vary depending on the species, understanding how changes in the bacterial community structure affect the host response to *Bd* infection may be key to successful *ex situ* programs. For example, *Becker et al. (2015)* showed that survival rates after exposure to *Bd* seem to be associated with the initial composition of the skin bacterial community, so different initial communities result in different outcomes for the host. Even though scientists have provided important insights on how different factors affect microbial assemblages in captive conditions (*Antwis et al., 2014*; *Michaels, Antwis & Preziosi, 2014*; *Michaels, Downie & Campbell-Palmer, 2014*), additional research is required to determine the impact of captivity on the proportion of bacteria with anti-*Bd* properties for specific amphibian species. Knowing how captive conditions affect the proportion and performance of bacteria with antifungal capacities will allow us to more accurately predict the fate of species considered for reintroduction into the wild.

Harlequin toads from the genus *Atelopus* are considered one of the most threatened groups of amphibians worldwide, with at least 71% of the species listed as critically endangered according to the *International Union for Conservation of Nature (2016)*. Although many species have been declining, some species from the lowlands still persist in their natural habitats (*Flechas, Vredenburg & Amézquita, 2015*). Other species, such as *A. elegans* from Ecuador, has only been found in one of its historical collection localities. Given the crisis of wild amphibian populations and the highly susceptibility of *Atelopus* to the fungal disease, conservation actions have been focused on keeping assurance colonies in order to protect these species. Conservation initiatives are committed to protecting those species facing the highest risk of extinction, and have included at least 15 species of the genus *Atelopus* in rescue programs (http://progress.amphibianark.org/model-programs). In this study, we evaluated the effect of *ex situ* conditions on bacteria with anti-*Bd* activity. To accomplish this, we performed antagonism assays to determine which bacterial species can inhibit *Bd* growth. Then we compared the activity between bacteria isolated from individuals in the wild with those maintained in captivity. We use culture-dependent methods to compare the skin bacterial community in three *Atelopus* species that persist in the wild but are also part of an *ex situ* breeding program. Our main goal was to evaluate the effect of captivity on the composition of the skin microbial community, focusing mainly in those that exhibit anti-*Bd* properties. Since captive environments have been described as less heterogeneous and less diverse compared with natural habitats, we hypothesized that animals in the wild would support a more diverse bacterial community and therefore a higher capacity to inhibit *Bd* growth, increasing chances of host survival.
**Table 1  Number of sampled individuals of each *Atelopus* species in wild and captive conditions.** The table includes the localities in Colombia and Ecuador where *Atelopus* individuals were sampled, the year when individuals were collected and entered into an *ex situ* facility, as well as the year when bacterial samples were taken for the analysis of the culturable portion of the skin microbiota.

| Species | Site of collection | Stage | Captive ind. | Year of collection | Date of sampling captive individuals | Wild ind. | Date of sampling wild individuals |
|---|---|---|---|---|---|---|---|
| *Atelopus elegans* | Esmeraldas, Ecuador | Juvenile | 2 | 1 ind - 2009 1 ind - 2010 | 2012 | – | – |
|  |  | Adult | 8 | 2 ind - 2009 3 ind - 2010 1 ind - 2011 2 ind - 2012 | 2012 | 5 | 2012 |
| *Atelopus* aff. *limosus* | Capurganá, Colombia | Juvenile | 2 | Born in captivity in 2011 | 2012 | – | – |
|  |  | Adult | 5 | 2 ind - 2008 3 ind -2009 | 2012 | 8 | 2009 |
| *Atelopus spurrelli* | Arusí, Colombia | Adult | 4 | 2011 | 2012 | 5 | 2009 |

## MATERIALS & METHODS

Research permits to conduct this study were provided by the Colombian National Parks Authority, the Ministerio de Ambiente de Colombia, and the Ministerio del Ambiente Ecuador (MAE) under permits DTSO 019-09, DTSO 001-09, No 10-07032012, 001-11 IC-FAU-DNB/MA and 11-2012-FAU-DPAP-MA.

### Study species

We studied three *Atelopus* species from the lowlands, two occurring in the Pacific coastal forests of Colombia and one in Ecuador (Fig. 1). We took skin bacterial samples of *A. spurrelli*, in the municipality of Arusí (5.57°N, 77.50°W; 90 m), *A.* aff. *limosus*, a species occurring near Capurganá (8.60°N, 77.33°W; 150 m) near the border between Colombia and Panama, and *A. elegans* from the Esmeraldas Province in northwestern Ecuador (1.04°N, 78.62°W; 265 m). In addition, we took skin swabs from captive *A. elegans*, *A.* aff. *limosus* and *A. spurrelli* from the *ex situ* programs in the Cali Zoo (Cali, Colombia) and the "Balsa de los Sapos" Conservation Initiative at the Pontificia Universidad Católica del Ecuador (Quito, Ecuador). Individuals in captivity came from the same populations where wild samples were collected (Table 1).

### *Ex situ* conditions

*Atelopus spurrelli* and *A.* aff. *limosus* were brought to the Cali Zoo facilities and kept isolated during quarantine (60 days). Then individuals were transferred to the enclosures (1 m × 0.5 m × 0.5 m) built using 6 mm-thick glass. To better simulate natural conditions, we created a 10 cm-depth pool that occupied a fifth of the tank's base. Each terrarium was connected to an external canister filter AE 306 Resun® (flow: 700L/H, power: 12 watts). The filters were placed below each terrarium and connected with $\frac{1}{2}$" tubing. Natural plants were placed in the tank to provide shelter, sleeping sites and visual barriers for the frogs, as well as to maintain high humidity, good water quality and low levels of nitrogenous waste. Bromeliads, aroids, mosses, orchids and calatheas were preferred because they tolerate high humidity levels. All individuals were fed five times a week with domestic crickets and fruit flies enriched with vitamin-mineral supplements. Because our study species occur in the

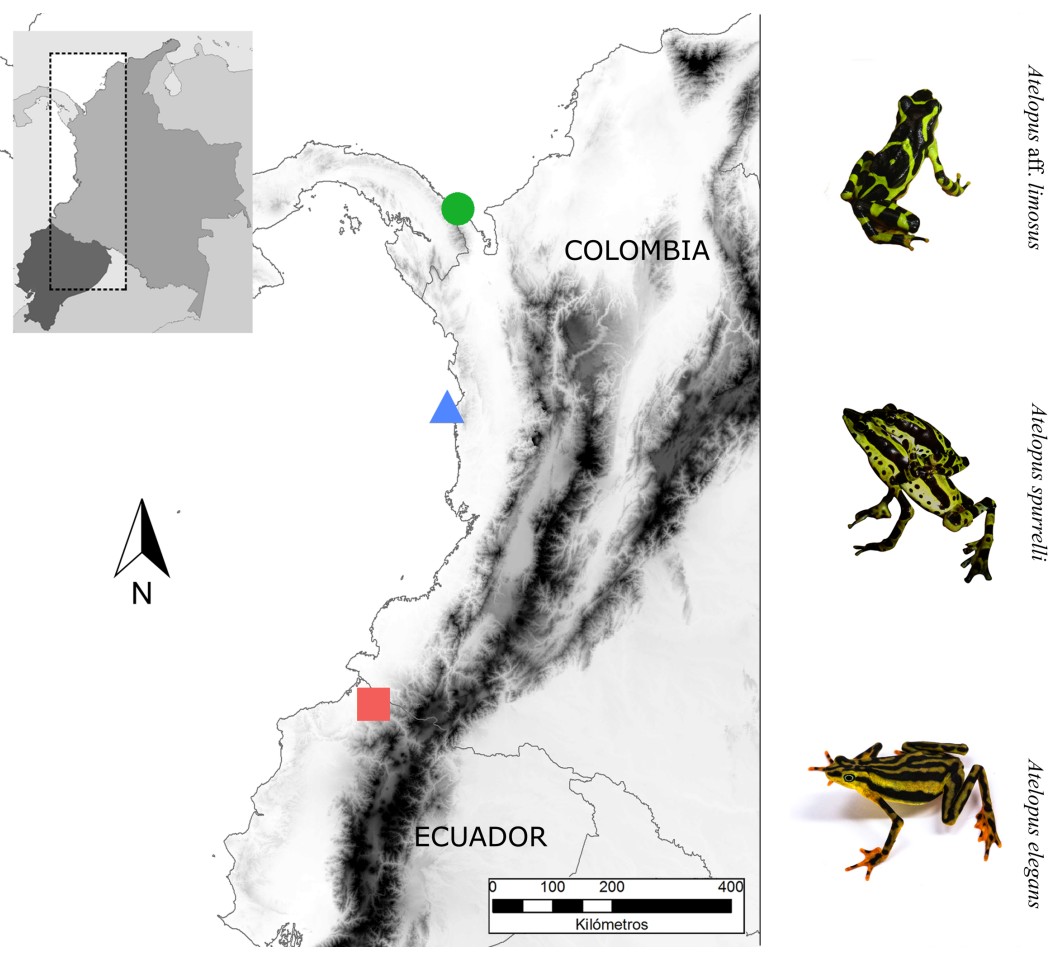

**Figure 1** **Sampling sites.** Map showing the localities where bacterial samples were obtained *Atelopus* aff. *limosus* (green dot), *Atelopus spurrelli* (blue triangle) and *Atelopus elegans* (red square). Individuals brought to captivity were collected in the same localities where samples from wild individuals were obtained. The map was created using the profiles provided by GADM.

lowlands of Colombia, we kept air temperature within enclosures between 24–30 °C and water temperature at 20.5–22 °C.

Individuals of *A. elegans* at ''Balsa de los Sapos'' Conservation Initiative were maintained isolated in independent containers during quarantine (60 days). Afterward, toads were transferred to plastic enclosures (30 cm × 20 cm × 20 cm) in which they were kept individually. Each terrarium contained natural plants, leaf litter, and a coconut bark to provide shelter and sleeping sites for the toads. Each container had 3 cm of water under false floor to maintain high levels of relative humidity. Clean water simulating artificial rain fell four times a day into each container during 5 min from a purification system with four filters: polyester film, UV, activated charcoal, and fabric filtration. All toads were fed three times a week with crickets that were sprinkled with vitamins and calcium supplement (Repashi Superfoods). Room temperature ranged between 19.6–22.9 °C.

## Bacterial isolation

In order to determine the effect of captive conditions on the composition of cutaneous bacteria, we took skin bacterial samples from captive animals: four *A. spurrelli*, seven *A.* aff. *limosus* and ten individuals of *A. elegans*. All bacterial swab samples from captive animals were collected in 2012 (Table 1). To compare bacterial communities of captive and wild frogs, samples from wild animals were taken from five individuals of *A. elegans* from Esmeraldas (Ecuador). For wild individuals of *A. spurrelli* and *A.* aff. *limosus* we used the published data from *Flechas et al. (2012)*.

Individuals were handled with new nitrile gloves and rinsed with 20 mL of sterile water to remove transient cutaneous microbiota. Each animal was sampled by running a sterile synthetic rayon swab (Medical Wire Equipment MWE 100) on their left, right of their body and ventral surfaces, hindlimbs, and interdigital membranes for a total of 50 strokes. Swabs were stored in 2 mL cryovials containing 1 mL DS solution, i.e., a weak salt solution resembling pond water (*Boyle et al., 2003*). Samples were refrigerated within 24 h and processed within 48 h after sampling. To isolate pure colonies, we followed the protocol described previously by *Flechas et al. (2012)*. Each isolate was cryopreserved in nutritive broth with 30% glycerol at $-80\,°C$.

## Growth inhibition assays

To determine the ability of each isolate to inhibit *Bd* growth we performed antagonism assays as previously published (*Harris et al., 2006*; *Lam et al., 2010*; *Flechas et al., 2012*). Assays were conducted using two *Bd* strains: JEL423 (provided by JE Longcore, University of Maine, Orono, USA) and EV001 (*Flechas et al., 2013*). Bacterial isolates from captive animals of the three *Atelopus* species and from wild *A. elegans* were challenged against both *Bd* strains in duplicate. Bacterial strains isolated from wild Colombian *Atelopus* were only tested against the strain JEL 423 since the Colombian strain was not available before 2012. Query bacteria that produced a clear halo were considered as inhibitory (see Fig. 1 in *Flechas et al., 2012*). Isolates that showed signs of growth inhibition were photographed under similar resolution and lighting conditions in a dark background in order to estimate antifungal ability.

Sections of approximately $3.0\,\mathrm{cm} \times 1.75\,\mathrm{cm}$ were taken from digital images of each culture plate: one section was taken from close to the bacterial growth (query bacteria) and two from sections were only *Bd* was present (control section of the plate). To estimate the antifungal action of bacteria, a graphical approach was developed using programming routines for R software (*R Development Core Team, 2016*). The three images were read into R using the function *raster* of 'raster' package (*Hijmans, 2015*) and then a frequency table was created based on the color present in each pixel using the function *freq* of the same package. We generated a continuous color scale ranging from 0 (white—representing only presence of *Bd*) to 255 (black—representing complete absence of *Bd*). To ensure these extreme values were present in the picture, two circles of equal dimensions (one of each color) were manually added to each image.

Because *Bd* does not grow uniformly over the plate's surface, two control assessments were employed: (a) We obtained kernel density estimates for each control picture and the

average of the modal color value from each picture was used in all posterior analyses. For this step, we used the function *density* from the 'stats' package in R (*R Development Core Team, 2016*) with its default arguments, and, (b) the frequency tables of each control picture were averaged by color, and then the color value corresponding to the mode of the kernel density estimates was used as control. These two control values were compared to the modal color value of the picture showing the bacterial effect on *Bd* growth. We assumed that lower experimental values than the control imply *Bd* inhibition (darker background in picture), and no differences between control and experimental values imply no antifungal effect (S1).

### Identification of bacterial isolates

DNA from pure cultures, including inhibitory and non-inhibitory strains were extracted as follows: one colony of each morphotype was re-suspended in 10 mL of pure water HPLC grade in a 0.2 mL PCR tube and boiled for 7 min at 95 °C. PCR amplifications were performed using universal primers 27F and 1492R (*Lane, 1991*) and the following parameters: an initial denaturation of 3 min at 95 °C, followed by 35 cycles of 45 s at 95 °C, 45 s at 52 °C and 90 s at 72 °C, and a final extension of 7 min at 72 °C. Each reaction consisted of 0.5 μL of each primer (1 μM), 3.0 μL of doubly distilled DNA-free water, 6 μL of GoTaq® Green Master Mix (1X; Promega) and 2 μL of the DNA extract. PCR results were checked by electrophoresis in 1% agarose gels. DNA sequences were cleaned and assembled using Geneious (*Kearse et al., 2012*). 16S rRNA sequences were then identified using BLASTn (*Altschul et al., 1990*) against the complete GenBank nucleotide database (http://www.ncbi.nlm.hih.gov) and the Greengenes database (http://greengenes.lbl.gov) using default parameters in both cases.

### Statistical analyses

In order to determine the variable (species or condition) that best explained the composition of the skin-associated bacterial community we used a Canonical Correspondence Analysis—CCA using the function *cca* in the 'vegan' package in R (*Oksanen et al., 2015*). In addition, to compare bacterial diversity between wild and captive animals, we counted the number of bacterial species in each category (species and condition) and then we performed a chi-square test using the 'stats' package in R (*R Development Core Team, 2016*). In order to determine the effect of captivity on bacterial assemblages, we used a chi-square test to check if there were differences in the proportion of bacterial species with antibacterial properties between conditions (wild vs. captive). Lastly, to test for differences in the capacity of bacterial species to inhibit *Bd* depending on their origin (wild vs. captive), we performed an ANOVA using the data obtained from the image analysis, after testing for normality and homogeneity of variances using Shapiro–Wilk and Bartlett tests, respectively.

## RESULTS

### Isolated bacteria from wild and captive individuals

Based on morphological characteristics, a total of 153 bacterial morphotypes were isolated from captive individuals: 70 from *A. elegans*, 62 from *A.* aff. *limosus*, and 21 from *A. spurrelli*. We recovered 40 morphotypes from wild *A. elegans*. Fifteen (21%) bacterial

**Table 2  Summary of the number of bacterial isolates recovered from three *Atelopus* species in captive and wild conditions.** The number in parenthesis represents the percentage of isolates with anti-*Bd* properties. In the last column we detailed the number of bacterial species that exhibit some degree of anti-*Bd* activity.

| Species | Condition | # of isolates | # Anti-*Bd* isolates | # Bacterial species with anti-*Bd* activity |
|---|---|---|---|---|
| *Atelopus elegans* | Captive | 70 | 15 (21%) | 7 from 3 genera |
| | Wild | 40 | 11 (27%) | 9 from 7 genera |
| *Atelopus aff. limosus* | Captive | 62 | 29 (47%) | 11 from 6 genera |
| | Wild | 77 | 20 (26%) | 7 from 3 genera |
| *Atelopus spurrelli* | Captive | 21 | 6 (28%) | 4 from 4 genera |
| | Wild | 21 | 6 (28%) | 5 from 2 genera |

morphotypes from captive and 11 (27%) from wild individuals of *A. elegans* inhibited *Bd* growth in the antagonism assays. For captive *A.* aff. *limosus* and *A. spurrelli*, 29 of 62 and six of 21 morphotypes inhibited *Bd* growth, respectively (Table 2). Since identification of bacteria based only on morphology is not completely reliable, we sequenced the 16S rRNA gene for all the isolated morphotypes. From 193 culturable morphotypes, we were able to identify bacteria representing 74 OTUs belonging to 37 genera. Through antagonism assays we found 28 OTUs (38%) in 13 genera showed some degree of anti-*Bd* activity. To define bacterial species we used a cut of 97% similarity. Sequences are available in GenBank with the following accession numbers: KY910042–KY910115 and KY938077–KY938172.

## Effect of captivity on the composition of skin bacterial community

Although captive and wild individuals from the same host species harbor cutaneous bacteria belonging to the same genera (Fig. 2), we found little overlap in bacterial species diversity between conditions (CCA: $\chi^2_{df=1} = 0.408$, $F = 1.23$, $P = 0.02$) (Table 3, Fig. 3). We did not find evidence for reduction in the number of bacterial species between wild and captive animals ($\chi^2_{df=2} = 3.6551$, $P = 0.1608$, Fig. 4). Antagonism experiments revealed that animals in the wild and individuals kept in *ex situ* facilities harbor in their skin bacteria with antifungal capacities. Differences in the proportion of bacteria with antifungal abilities between captive (22/64) and wild individuals (18/53) were not significant (Fisher exact test, $P = 1$, Odds ratio = 1.01, Fig. 4).

## Anti-*Bd* activity in shared bacterial strains

To determine if anti-*Bd* activity changes depending on the origin of the bacteria, we compared the ability to inhibit *Bd* only for those bacterial species that were found in both captive and wild animals. We detected five bacterial species that inhibited *Bd* growth in both conditions including: *Acinetobacter* sp., *Chryseobacterium* sp., *C. meningosepticum*, *Pseudomonas putida*, and *Pseudomonas* sp. The inhibition was measured as a reduction of density of *Bd* colonies (based on the presence of darker pixels in the experimental area). Differences in the extent of inhibition between bacteria isolated from wild and captive animals were not detected ($F = 0.503$, $P = 0.73$; Fig. 5). Bacteria inhibit *Bd* growth irrespective of the *Bd* strain used (JEL or EV001, Fisher exact test, $P = 0.051$, Odds ratio = 0.385).
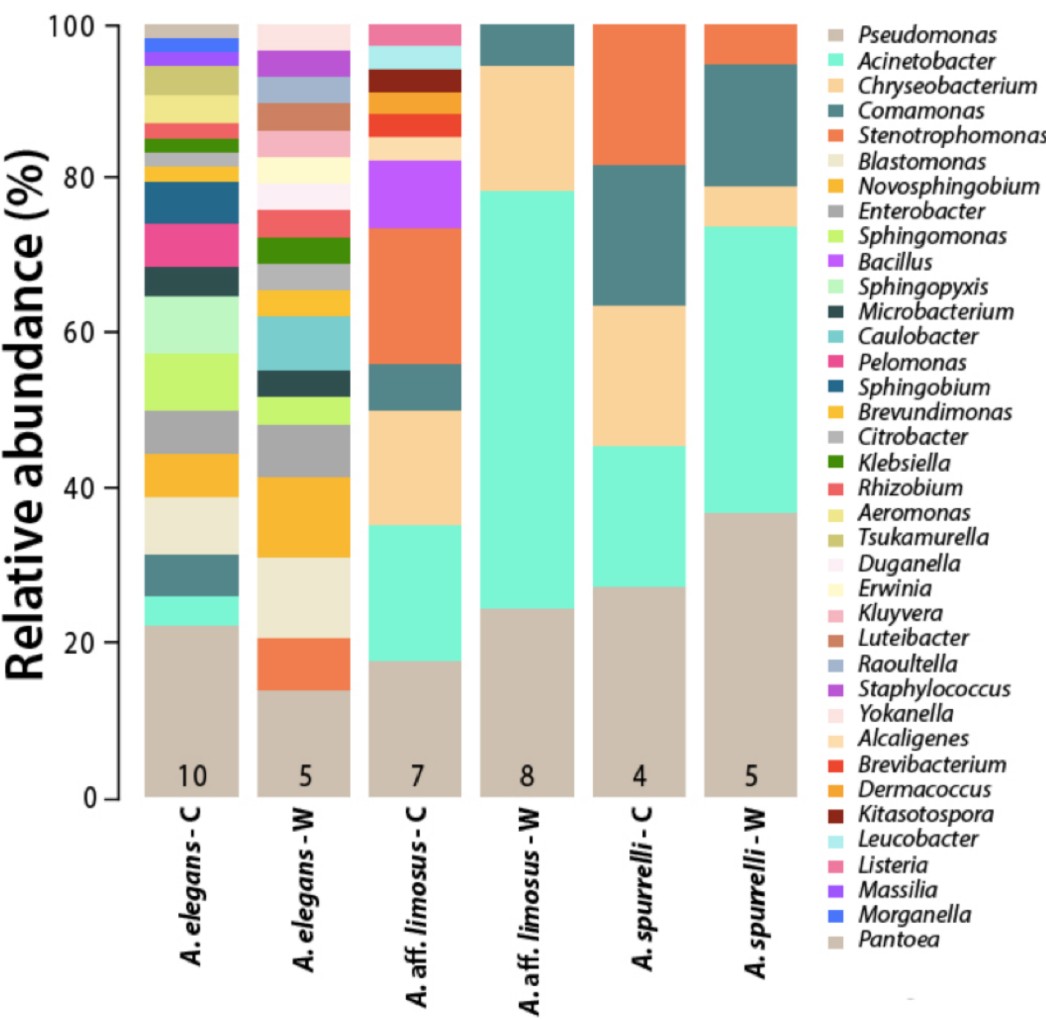

**Figure 2** **Differences in the relative abundance of culturable bacterial genera isolated from the skin of captive and wild individuals in three species of *Atelopus*.** Bars are grouped by species and condition. Numbers inside the bars denote the number of toads sampled per category. C, Captive; W, Wild.

## DISCUSSION

The genus *Atelopus* is one of the most threatened groups of amphibians worldwide. Although more than 70% of the species in the genus are listed as critically endangered according to the *International Union for Conservation of Nature (2016)*, some species have persisted in the wild (*Flechas, Vredenburg & Amézquita, 2015*). In other cases, such as *A. elegans* from Ecuador, this species has been recently rediscovered but only in one of the historical localities. Given the high susceptibility of the genus to chytridiomycosis, and the lack of a cure that could allow them to survive in nature, conservation actions have been focused on keeping assurance colonies. In this study, we describe culturable bacterial assemblages for individuals of three *Atelopus* species that remain in the wild, in apparently healthy conditions, and individuals (from the same species and collection localities) brought into captivity. This study sought to determine the effect of captive conditions

**Table 3  Bacterial species isolated from the skin of *Atelopus elegans*, *Atelopus* aff. *limosus* and *Atelopus spurrelli*.** C, Captive; W, Wild. The number in parenthesis represents the number of individuals sampled in each condition. Numbers in each cell represent the number of isolates for each bacterial species. Species that showed anti-*Bd* activity are marked with an asterisk (*). Bacteria highlighted in grey correspond to those present in captive and wild animals.

| Family | Bacterial isolates | *A. elegans* | | *A.* aff. *limosus* | | *A. spurrelli* | |
|---|---|---|---|---|---|---|---|
| | | C (10) | W (5) | C (7) | W (8) | C (4) | W (5) |
| Moraxellaceae | *Acinetobacter baumannii* | – | – | 1 | 3 | – | 2 |
| Moraxellaceae | *Acinetobacter bereziniae* | – | – | 1 | – | – | – |
| Moraxellaceae | *Acinetobacter calcoaceticus* | – | – | 1 | 2 | – | 1 |
| Moraxellaceae | *Acinetobacter genomosp.* | – | – | – | 1 | – | – |
| Moraxellaceae | *Acinetobacter haemolyticus* | – | – | – | – | – | 1* |
| Moraxellaceae | *Acinetobacter gyllenbergii* | – | – | – | 5* | – | 1* |
| Moraxellaceae | *Acinetobacter junii* | – | – | – | 1 | – | – |
| Moraxellaceae | *Acinetobacter* sp. | 2 | – | 3* | 7* | 2 | 2* |
| Moraxellaceae | *Acinetobacter venetianus* | – | – | – | 1 | – | – |
| Aeromonadaceae | *Aeromonas hydrophilia* | 1 | – | – | – | – | – |
| Aeromonadaceae | *Aeromonas* sp. | 1 | – | – | – | – | – |
| Alcaligenaceae | *Alcaligenes faecalis* | – | – | 1* | – | – | – |
| Bacillaceae | *Bacillus* sp. | – | – | 1 | – | – | – |
| Bacillaceae | *Bacillus cereus* | – | – | 2* | – | – | – |
| Sphingomonadaceae | *Blastomonas natatoria* | 4 | 3* | – | – | – | – |
| Brevibacteriaceae | *Brevibacterium aureum* | – | – | 1 | – | – | – |
| Caulobacteraceae | *Brevundimonas aurantiaca* | 1 | 1 | – | – | – | – |
| Caulobacteraceae | *Caulobacter crescentus* | – | 1 | – | – | – | – |
| Caulobacteraceae | *Caulobacter vibroides* | – | 1 | – | – | – | – |
| Flavobacteriaceae | *Chryseobacterium indologenes* | – | – | 2* | – | – | – |
| Flavobacteriaceae | *Chryseobacterium meningosepticum* | – | – | 1* | 1* | 2* | 1 |
| Flavobacteriaceae | *Chryseobacterium* sp. | – | – | 2* | 5* | – | – |
| Enterobacteriaceae | *Citrobacter freundii* | – | 1* | – | – | – | – |
| Enterobacteriaceae | *Citrobacter* sp. | 1* | – | – | – | – | – |
| Comamonadaceae | *Comamonas* sp. | 3 | – | | 1 | 1 | 1 |
| Comamonadaceae | *Comamonas testosteroni* | – | – | 2 | 1 | 1* | 2 |
| Oxalobacteraceae | *Duganella* sp. | – | 1 | – | – | – | – |
| Dermacoccaceae | *Dermacoccus* sp. | – | – | 1 | – | – | – |
| Enterobacteriaceae | *Enterobacter asburiae* | 1* | – | – | – | – | – |
| Enterobacteriaceae | *Enterobacter* sp. | 2 | 2 | – | – | – | – |
| Enterobacteriaceae | *Erwinia amylovora* | – | 1 | – | – | – | – |
| Streptomycetaceae | *Kitasatospora phosalacinea* | – | – | 1 | – | – | – |
| Enterobacteriaceae | *Klebsiella oxytoca* | 1 | – | – | – | – | – |
| Enterobacteriaceae | *Klebsiella* sp. | – | 1 | – | – | – | – |
| Enterobacteriaceae | *Kluyvera ascorbata* | – | 1* | – | – | – | – |
| Microbacteriaceae | *Leucobacter* sp. | – | – | 1 | – | – | – |
| Listeriaceae | *Listeria* sp. | – | – | 1 | – | – | – |
| Xanthomonadaceae | *Luteibacter rhizovicinus* | – | 1* | – | – | – | – |

**Table 3** (*continued*)

| Family | Bacterial isolates | *A. elegans* | | *A.* aff. *limosus* | | *A. spurrelli* | |
|---|---|---|---|---|---|---|---|
| | | **C (10)** | **W (5)** | **C (7)** | **W (8)** | **C (4)** | **W (5)** |
| Oxalobacteraceae | *Massilia* sp. | 1 | – | – | – | – | – |
| Microbacteriaceae | *Microbacterium foliorum* | 1 | – | – | – | – | – |
| Microbacteriaceae | *Microbacterium* sp. | 1 | – | – | – | – | – |
| Microbacteriaceae | *Microbacterium testaceum* | – | 1 | – | – | – | – |
| Enterobacteriaceae | *Morganella morganii* | 1 | – | – | – | – | – |
| Sphingomonadaceae | *Novosphingobium aromaticivorans* | 1 | – | – | – | – | – |
| Sphingomonadaceae | *Novosphingobium subterraneum* | 2 | **3\*** | – | – | – | – |
| Enterobacteriaceae | *Pantoea agglomerans* | 1 | – | – | – | – | – |
| Comamonadaceae | *Pelomonas puraquae* | 3 | – | – | – | – | – |
| Pseudomonadaceae | *Pseudomonas aeruginosa* | **1\*** | – | – | – | – | – |
| Pseudomonadaceae | *Pseudomonas fulva* | 1 | **1\*** | – | – | – | – |
| Pseudomonadaceae | *Pseudomonas geniculata* | – | – | **1\*** | – | – | – |
| Pseudomonadaceae | *Pseudomonas montielli* | – | – | – | – | **1\*** | – |
| Pseudomonadaceae | *Pseudomonas mosselii* | **1\*** | – | **1\*** | – | – | – |
| Pseudomonadaceae | *Pseudomonas plecoglossicida* | – | – | 1 | – | – | **1\*** |
| Pseudomonadaceae | *Pseudomonas putida* | **2\*** | **1\*** | **2\*** | 1 | 1 | **2\*** |
| Pseudomonadaceae | *Pseudomonas saccharophilia* | 1 | – | – | – | – | – |
| Pseudomonadaceae | *Pseudomonas* sp. | **4\*** | **2\*** | 2 | 1 | 1 | 1 |
| Pseudomonadaceae | *Pseudomonas nitroreducens* | – | – | – | – | – | 1 |
| Pseudomonadaceae | *Pseudomonas straminea* | – | – | – | – | – | 1 |
| Pseudomonadaceae | *Pseudomonas tolaasii* | – | – | – | 1 | – | 1 |
| Pseudomonadaceae | *Pseudomonas veronii* | – | – | – | **6\*** | – | – |
| Pseudomonadaceae | *Pseudomonas vranovensis* | **2\*** | – | – | – | – | – |
| Enterobacteriaceae | *Raoultella ornithinolytica* | – | 1 | – | – | – | – |
| Rhizobiaceae | *Rhizobium* sp. | 1 | 1 | – | – | – | – |
| Sphingomonadaceae | *Sphingobium yanoikuyae* | 2 | – | – | – | – | – |
| Sphingomonadaceae | *Shipngomonas paucimobilis* | 1 | – | – | – | – | – |
| Sphingomonadaceae | *Sphingomonas* sp. | 4 | 1 | – | – | – | – |
| Sphingomonadaceae | *Sphingopyxis* sp. | 4 | – | – | – | – | – |
| Staphylococcaceae | *Staphylococcus capitis* | – | 1 | – | – | – | – |
| Xanthomonadaceae | *Stenotrophomonas maltophilia* | – | 1 | **4\*** | – | 1 | 1 |
| Xanthomonadaceae | *Stenotrophomonas* sp. | – | 1 | **2\*** | – | **1\*** | – |
| Nocardiaceae | *Tsukamurella* sp. | 1 | – | – | – | – | – |
| Nocardiaceae | *Tsukamurella tyrosinosolvens* | 1 | – | – | – | – | – |
| Enterobacteriaceae | *Yokenella regensburgei* | – | 1 | – | – | – | – |

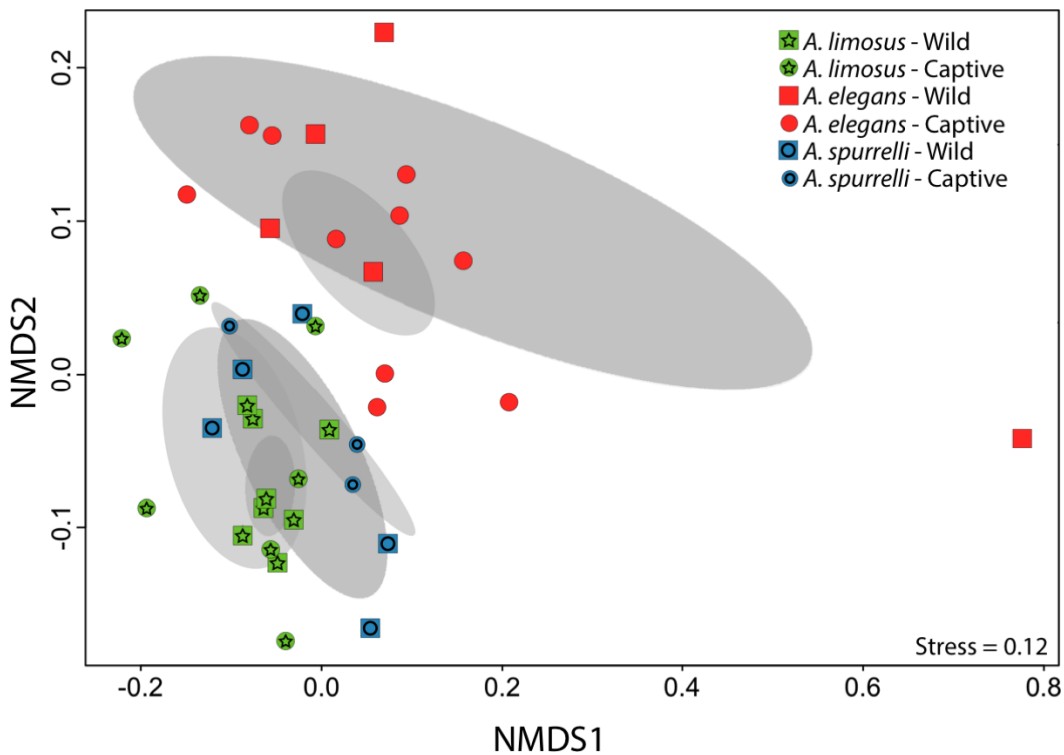

**Figure 3** **Culturable bacterial communities between captive and wild individuals of *Atelopus* exhibit little overlap.** Non-metric Multidimensional Scaling (NMDS) ordination of Bray–Curtis distances between microbial communities. Each symbol represents one sampled individual. Ellipses represent 95% confidence intervals (Dark grey, wild; light grey, captive).

on the anti-*Bd* bacterial community in one of the most threatened genera worldwide (*La Marca et al., 2005*; *Mendelson et al., 2006*). Our main goal was not only to describe the culturable portion of the bacterial community for each condition, but the effect of captivity on the proportion of beneficial bacteria and their performance as *Bd* inhibitors. Although our results show that captive individuals harbor bacteria with anti-*Bd* properties, further long-term research is key to evaluate the interactions between the skin microbial communities and their amphibian hosts since those can differ from one population to the next even if the bacterial community composition is similar (*Rebollar et al., 2016*).

Several studies have reported changes in the microbial associated communities when animals are moved from the wild to captive environments (*Isaacs et al., 2009*; *Dhanasiri et al., 2011*; *Wienemann et al., 2011*; *Nelson et al., 2013*). In amphibians, research assessing the effect of captivity on bacterial communities is increasing (*Antwis et al., 2014*; *Becker et al., 2014*; *Michaels, Antwis & Preziosi, 2014*; *Kueneman et al., 2016*). Through culture-independent and dependent approaches, these studies have demonstrated changes in the skin bacterial community composition due to conditions imposed by captivity. Our results suggest that *ex situ* conditions, at least for these three *Atelopus* species, did not reduce the diversity of bacteria or the proportion of anti-*Bd* bacteria, as determined by culture-based assays. Moreover, we found a higher number of culturable bacterial species in captive

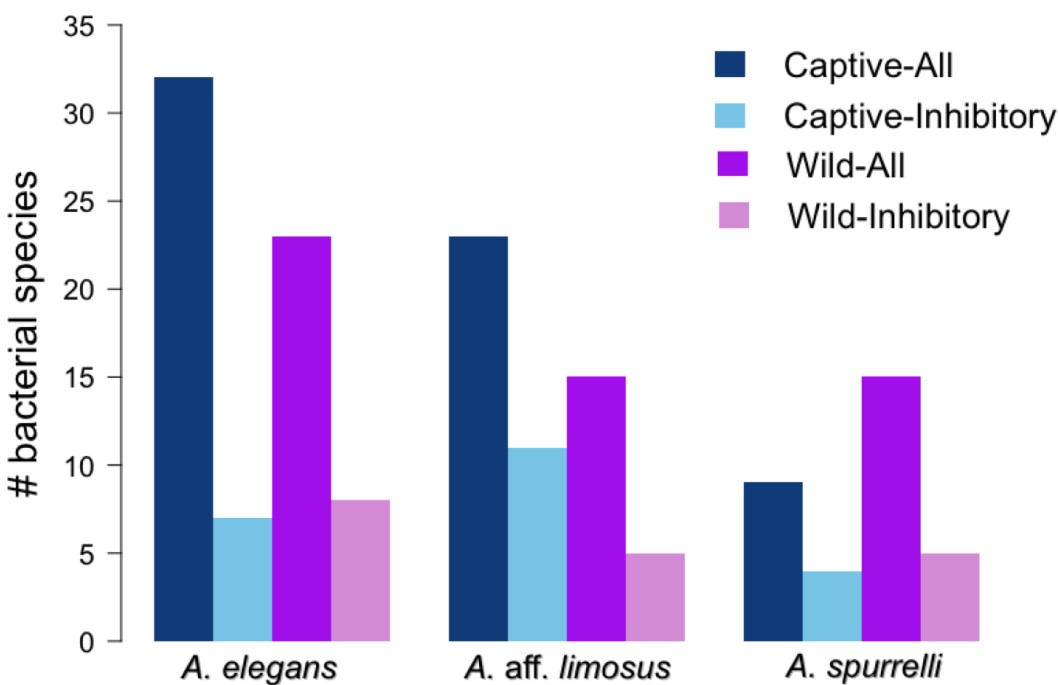

**Figure 4** *Atelopus* **in the wild and in captivity harbor in their skin culturable bacteria with antifungal abilities.** Darker bars show the total number of isolated bacteria in each condition (wild vs. captive). Lighter bars represent the number of bacterial species with anti-Bd activity in each condition (wild vs. captive).

animals compared to wild individuals. Our findings are consistent with those obtained for other vertebrates including birds and mammals (*Xenoulis et al., 2010*; *Nelson et al., 2013*), where gut microbial diversity was higher in captive animals, and are also consistent with the findings for the Panamanian golden frogs (*A. zeteki*) where OTU richness and phylogenetic diversity was higher in captive frogs (*Becker et al., 2014*). We found that species of bacteria belonging to the genera *Pseudomonas, Chryseobacterium* and *Acinetobacter* isolated from wild and captive animals inhibited *Bd* in the challenge assays. In a recent study, bacterial isolates from these three groups were also reported as *Bd* inhibitors and they appeared to be widely distributed. The proportion of bacteria that showed anti-*Bd* activity for those genera was 73% for *Pseudomonas,* 55% for *Chryseobacterium* and 37% for *Acinetobacter* (*Woodhams et al., 2015*).

Variations in bacterial communities due to captive conditions have been demonstrated in red-backed salamanders (*Plethodon cinereus*), where environmental microbes seemed to affect host-microbial diversity and appeared to be important as regulators of the host core community (*Loudon et al., 2014*). In terms of their associated microbiota, captive environments appear to be less diverse and less heterogeneous than wild habitats, which might imply that host associated bacterial communities for animals in *ex situ* facilities should be less diverse (*Sabino-Pinto et al., 2016*). The enclosures where frogs are maintained provide a modified environment that can also vary depending on the zoo or facility. These conditions could facilitate the transmission of new bacteria capable

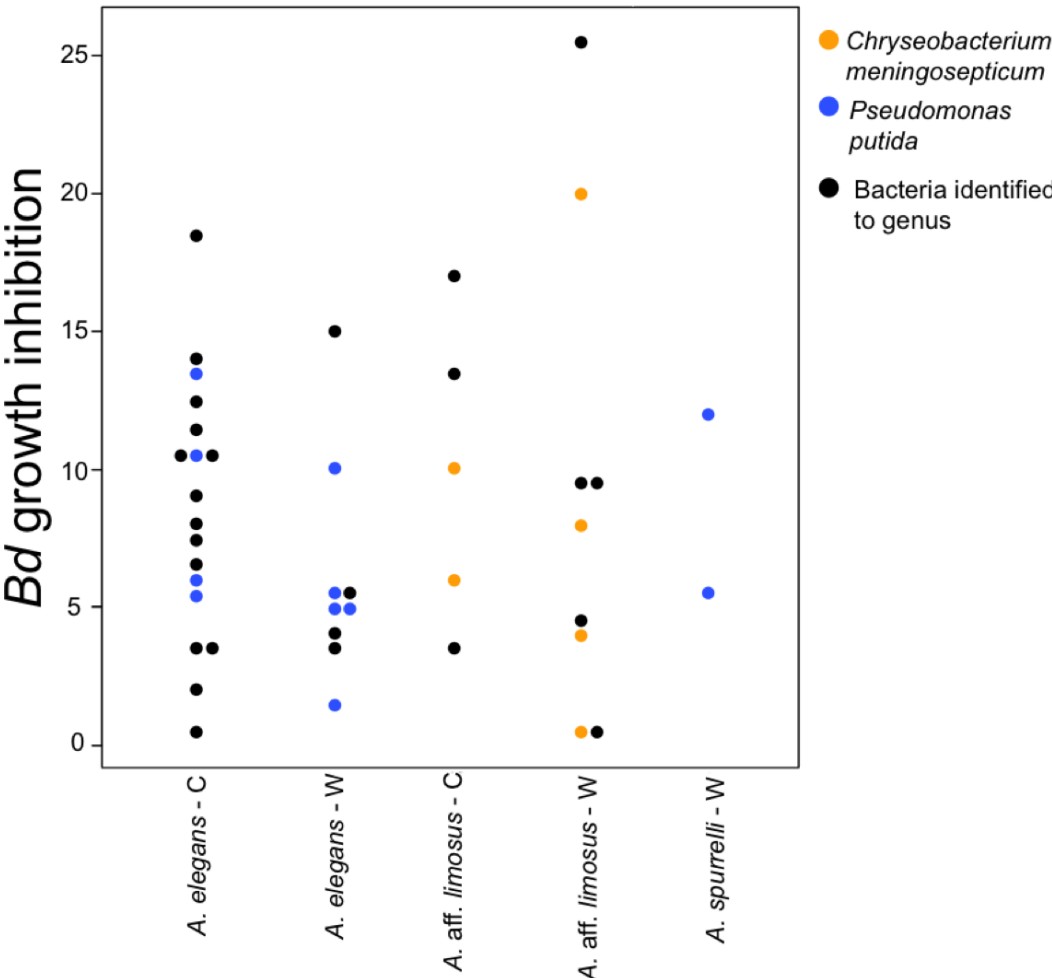

**Figure 5** **Bacteria isolated from wild and captive animals did not differ in the extent of *Bd* inhibition.** Colored dots were only used for those bacteria that were identified to species. Black dots correspond to the bacteria that were identified to genus.

of colonizing the frog skin, as suggested for other animals in captivity (*Nelson et al., 2013*). We found that culturable bacterial communities from captive animals at the Cali Zoo (*A.* aff. *limosus* and *A. spurrelli*) were not the same but more similar to each other than compared with *A. elegans*. The slight difference between captive *A.* aff. *limosus* and *A. spurrelli* (Fig. 2), suggests that a "host" component is also playing a role shaping skin bacterial communities, as suggested previously for Japanese amphibians (*Sabino-Pinto et al., 2016*). Although the cages used in this study simulated natural conditions, we propose that the observed differences between frogs from both *ex situ* facilities might be related to the environmental conditions inside the enclosures (i.e., substrate and water sources). Although, there is no solid evidence on which mechanisms are the frogs using to acquire bacteria from the environment, and which are the key factors modeling bacterial assemblages, we suggest that either the environment or host characteristics are modeling bacterial communities in these species of harlequin toads.

A cure or mechanism that allows threatened amphibian species to survive chytridiomycosis in their natural habitats is still needed. Although many different disease mitigation strategies have been proposed (*Woodhams et al., 2011*; *Bosch et al., 2015*), keeping viable assurance colonies in captivity seems to be the best option for *Bd*-susceptible species yet (*Gratwicke et al., 2015*). The main goal of *ex situ* programs is to eventually reintroduce animals to their natural habitats. Management in captivity can lead to alterations of the species' microbiota that may reduce survival chances after reintroduction (*Redford et al., 2012*). In this study, we isolated a similar number of bacterial species with anti-*Bd* activity from captive and wild individuals of three *Atelopus* species, suggesting that they might count on their skin bacteria as a potential defense barrier against *Bd*. *Redford et al. (2012)* proposed that more studies are needed to determine changes and alterations in the animals' microbiome in order to increase the success of reintroduction efforts. Here, we presented evidence that although bacterial community structure changed in captivity, the predicted anti-*Bd* function remained intact. In addition, it seems that animals are acquiring new microorganisms that are also providing protection against the fungal pathogen. Our findings suggest that selection acted on bacterial community function (e.g., antifungal activity and protection) as opposed to selection for specific community structures. We hope this study might be considered when reintroduction plans are proposed as the next step for many species that are now part of *ex situ* programs.

## ACKNOWLEDGEMENTS

For assistance in the field we thank D Velalcázar, A Barahona, J Méndez and C Silva. We would like to thank "Balsa de los Sapos" Conservation Initiative and the Cali Zoo for allowing us to take samples of captive animals. We thank B Liger for creating the distribution map. We thank AJ Crawford for helpful comments and suggestions that greatly improved this manuscript. We would like to thank the reviewers for their valuable comments on the manuscript.

### Funding

This work was supported by the Association of Zoos and Aquariums Conservation Endowment Fund (08-836), the Facultad de Ciencias at Universidad de los Andes, and the Dirección General Académica of the Pontificia Universidad Católica del Ecuador. The funders had no role in study design, data collection and analysis, decision to publish, or preparation of the manuscript.

### Grant Disclosures

The following grant information was disclosed by the authors:
Association of Zoos and Aquariums Conservation Endowment Fund: 08-836.
Facultad de Ciencias at Universidad de los Andes.
Dirección General Académica of the Pontificia Universidad Católica del Ecuador.

## Competing Interests

The authors declare there are no competing interests.

## Author Contributions

- Sandra V. Flechas conceived and designed the experiments, performed the experiments, analyzed the data, contributed reagents/materials/analysis tools, wrote the paper, prepared figures and/or tables, reviewed drafts of the paper.
- Ailin Blasco-Zúñiga conceived and designed the experiments, performed the experiments, analyzed the data, wrote the paper, prepared figures and/or tables, reviewed drafts of the paper.
- Andrés Merino-Viteri analyzed the data, contributed reagents/materials/analysis tools, wrote the paper, prepared figures and/or tables, reviewed drafts of the paper.
- Valeria Ramírez-Castañeda performed the experiments, reviewed drafts of the paper.
- Miryan Rivera contributed reagents/materials/analysis tools, reviewed drafts of the paper.
- Adolfo Amézquita conceived and designed the experiments, contributed reagents/materials/analysis tools, reviewed drafts of the paper.

## Animal Ethics

The following information was supplied relating to ethical approvals (i.e., approving body and any reference numbers):

Universidad de los Andes approved this research.

## Field Study Permissions

The following information was supplied relating to field study approvals (i.e., approving body and any reference numbers):

Research permits were provided by the Colombian National Parks authority, the Ministerio de Ambiente de Colombia, and the Ministerio del Ambiente Ecuador (MAE) under permits DTSO 019-09, DTSO 001-09, No 10-07032012, 001-11 IC-FAU-DNB/MA and 11-2012-FAU-DPAP-MA.

## Data Availability

Sequences are available in GenBank with the following accession numbers: KY910042–KY910115 and KY938077–KY938172.

## Supplemental Information

Supplemental information for this article can be found online at http://dx.doi.org/10.7717/peerj.3594#supplemental-information.

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
