# Peer review of "The effect of captivity on the skin microbial symbionts in three Atelopus species from the lowlands of Colombia and Ecuador"

_PeerJ, doi:10.7717/peerj.3594_

## Round 0.1 · original submission · Major Revisions

Having looked at the manuscript and received 3 reviews I have some concerns particularly around the framing of the paper and interpretation of the results. I do not believe these are difficult to correct but may take some time therefore I've suggested Major Corrections. I look forward to seeing these in the near future.

Reviewer 1 ·

Basic reporting

I made suggestions on grammar.

Experimental design

OK, although sample sizes are a bit low.

Validity of the findings

The overall conclusion is solid.

Additional comments

This is an interesting study that compares anti-Bd bacterial isolates from frogs in nature with frogs in captivity. Some of the comparisons are based on previously published data from a study that used different conditions in captivity. However, the overall result is that frogs had anti-Bd isolates in captivity to the same degree as in nature. Thus, there appeared to be no reduction in “defensive function” for frogs in captivity. Of course, for these frogs, anti-Bd function needs to be enhanced for them to survive in nature, but at least anti-Bd function does not seem to decline in captivity.

Were the captive animals collected from the same populations as the populations sampled in nature?
The authors can highlight the point that although bacterial community structure changed in captivity, the predicted anti-Bd function remained intact. This suggests that selection has acted on bacterial community function (eg., antifungal activity and protection) as opposed to selection for specific community structures.

In Figure 2, at least for limosus and spurrelli, it appears that there is considerable overlap in bacterial species. There is lower diversity on the wild animals, but those OTUs appear to be present in captivity. In line 220, I suggest a rewording. There was actually high overlap, by the alpha and beta diversity differed.

More details on the conditions in captivity would be helpful (L 119). How were the conditions different for the different frog species?

The Discussion section can be shortened. For example, the paragraph beginning on line 287 is overly speculative.

A key reference to consider is Loudon et al. (ISME J, 8:830-840), which investigated what happened to amphibian skin microbial communities under two conditions in captivity.

Other comments:

Abstract line 2: change to “safeguarded”
Abstract: change to “Through sequencing of the 16S rRNA gene”
L 26: “have become more problematic”
27: delete “Only”
33: delete the comma
36: delete “basically”
50: change to “Belden & Harris”
58: delete the comma
60: delete the comma after “and”
62: begin a new paragraph with “Cutaneous microbes”
81: delete the comma
98: change the comma to a period
99: delete the comma
133: change “manipulated” to “handled”
134: use “microbiota” instead of “microflora” throughout the manuscript
154: move “approximately” to before 3.0
193: change “are” to “were”
195: delete the comma
196: Use past tense “were” though out Methods and Results
200: Were the assumptions of ANOVA satisfied?
215: delete the comma
223: change “quantity” to “number”
226: number or proportion? I think it is proportion if you are using a chi-square test.
240: Fig. 5?
251: change to “genera”
252: Change to “Even though bacterial assemblages…”
257: delete the commas
264: change to “ex situ conditions did not reduce”
284: there are several studies that show a host component to amphibian skin bacterial community structure while holding environment constant (e.g., McKenzie et al. 2012. ISME J)
289: add references after “individuals”. Also, please explain “larger differences between species”. Larger than what? What is the comparison?
300: change “which” to “what”
305: delete the comma
307: for species endangered by Bd.
Fig. 3: The ellipses are not very informative. Also, why 5 ellipses and not 6?
The Figure Legends should present the result. For example, in Fig. 1, species overlap was minimal between captive and wild individuals.

Reviewer 2 ·

Basic reporting

The manuscript could do with further checks for grammar and punctuation. Some sections are better than others (e.g. methods).

Many statements are unclear and require rewording to improve grammar and/or remove ambiguity.

The interpretation of Bd and its global effects could be more concise. Further references are suggested in the reviewed doc.

Figures are not of sufficient quality for publication due to the resolution of the image files. In-figure text appears fuzzy and as if it has bled into the page; data points in Fig. 5 are blotches, and those of 'A. aff. limosus - W' appear to be a different size.

Experimental design

Specific comments are on the attached document.

Validity of the findings

My main concern regarding this manuscript is the interpretation of results in the last paragraph of the discussion. Statements such as "...allow us to predict that these individuals will be protected against the pathogen if released in the wild" are at best misplaced, at worst dangerous. On no account should reintroduction of amphibians be encouraged without knowing detailed information on the complex relationship between host, pathogen, microbiota, and ecology, which can be dictated by not only the bacterial composition on individual, and within populations of amphibians, but also by the interactions between and within amphibian and microbial communities, which themselves can differ from one population to the next, even if the species composition does not alter.

The presence of Bd-resistant bacteria and/or anti-Bd activity does not confer host, species, or population resistance.

Additional comments

This is an interesting study that sheds light on the differences that may be observed in skin microbiota between captive and wild amphibians. However, the manuscript would benefit from a re-framing of perspective when in comes to interpreting the results, and from exploring the meaning and validity of the observed variation.

General comments are in the attached document.

Annotated reviews are not available for download in order to protect the identity of reviewers who chose to remain anonymous.

Reviewer 3 ·

Basic reporting

This is an interesting and important study that provides important new data on our understanding of the role of cutaneous bacteria on the ecology and conservation of amphibians. The data have been obtained using appropriate methods, analyzed using appropriate procedures, and presented in an overall clear way.

Accession numbers of the newly sequenced isolates are not given. This is an absolute must and needs to be integrated in the next version of the manuscript. Please submit all sequences to Genbank, mention accession numbers in the main manuscript and include a detailed table with accession numbers as supplementary material.

Otherwise I almost only have minor corrections and suggestions for revision, but of these I have many. The authors should pay attention to the style and logic of writing, typos, and especially their use of tenses – they constantly switch between past and present tense when describing their results, and need to stick to one of them (preferably past tense which seems to be what they mostly use.
I will not make the standard suggestion of “proofreading by a native speaker here” – although the authors are not native speakers, I know that several of them are perfectly able to phrase their text into correct English. All that is needed here is a bit more effort and careful proofreading before resubmission.

In addition, I feel the authors should invest some effort to explain more stringently and more clear their approach and their methods. It took me some time to understand what they did, with constant flipping forth and back between tables, figures and text. I do not mean to say that a lot of additional words are needed. But for instance:
- it would be good in the figure caption to repeat again how these data were obtained (some wording such as “culture-based” or “cultured isolates” would already help.
- in the first section of the Results, it would be good to include some summary statistics also about the genera (to how many bacterial “species” =OTUs did the cultured bacteria belong and to how many genera (and the number of genera then should probably agree with those in Fig. 2, if I interpreted that graph correctly.

Finally, one somewhat major point is the fact that one important paper (Woodhams, D. C. et al. 2015 Antifungual isolates database of amphibian skin-associated bacteria and function against emerging fungal pathogens. Ecology 96, 595.) is not cited here. I strongly suggest the authors make an effort to integrate not only this paper, but to compare their data with that database (number of inhibitory isolates per genus / how many of their antifungal bacteria species are already includedin the database and were they also in previous studies identified as inhibitory? And so on. I would even suggest that the results of this study should also be submitted to that data base. At least, the authors should present the original data of their inhibition experiments for each strain in a supplementary table, making future integration into the Woodhams database easier, and thereby making their data more directly available to the scientific and conservation community.

The following is a series of minor points for revision, which however is not complete. Please check the entire text for similar issues and correct these as well!

Abstract: “it is still unknown how captivity affects amphibian beneficial bacteria”
This statement is a bit too strong given that several studies have already dealt with captivity effects on amphibian cutaneous bacteria, even if these studies did not specifically targeted “beneficial” bacteria. Try to rephrase, “unknown” is a very strong word and maybe can be replaced. Maybe something like: While previous studies have addressed captivity effects ... it remains poorly studied whether and how captivity effects the proportion of beneficial bacteria on amphibian hosts. “

Abstract: Please at first mention of Atelopus, make sure you say that this is an amphibian.

Line 46: Why “furthermore”? Please rephrase.

80 better use USA instead of US

84, Anaxyrus

85 assemblages’ --> assemblage

85 Why do we “need” to do this? Rephrase!

87 seemed

97 “To our knowledge this is the first study” – I strongly discourage such statements which some journals (e.g. PNAS) do not allow at all. Your study is interesting in itself, so there is no need to emphasize how much it is “the first” to do anything.

105 maintained

125-127 It is hard to understand why you here repeat information on “bacterial sampling”. I first thought in both cases you were referring to a different set of samples. Please streamline these two paragraphs and avoid any repetitive sentences.


162. I think “ensured” is a more appropriate term here

188-190. I think the first sentence is not really needed here. By now it should be clear that your approach is culture-based.


215 belonged

209-216. As I have already mentioned above, this section lacks clarity. You first speak about morphotypes, then about species, but it is not clear how either was identified. Were morphotypes just determined from the plates, and then you sequenced the isolates and determined how many species there are? How did you define a bacterial species (<3% 16S divergence?). Please also give information on the genera here so that this information here can be matched e.g. with Fig. 2. In general I think that this section should be completely rewritten.

223 evidence for

228 were --> was

229 different between what? unclear.

247 For someone not familiar with Atelopus this is not clear: Apparently the three species survive in the wild despite Bd ... so, why have any individuals of the same species been included in captive programs as “emergency measure”? Apparently there was no emergency here? This should be explained more clearly.

287 This seems to be the only place in the paper where you acknowledge that your study only targeted a small part of the skin microbial community. Although Walke et al. demonstrated that many cutaneous bacteria of amphibians can be cultured (Walke JB, Becker MH, Hughey MC, Swartwout MC, Jensen RV, Belden LK (2015) Most of the dominant members of amphibian skin bacterial communities can be readily cultured. Applied and Environmental Microbiology 81: 6589-6600.) it is clear that in your approach, you have only been able to isolate a fraction of the OTUs that are present on your toads (as obvious from NGS-amplicon-based approaches that regularly identify many more OTUs). Acknowledging this uncertainty is important, aand this should happen much earlier in the Discussion. Statements such as “Our results showed that ex situ conditions are not reducing the diversity of skin bacteria or the proportion of anti-Bd bacteria;” are clearly overstatements and must be avoided. You found indications for this, true, but you cannot make a definite statement here. So, please frame this using “might”, or “indicate” or “Our data allow to hypothesize” or similar phrasing. Same in line 311 where you could simply say “we isolated a similar number” instead of “harbor almost the same”

300 “using” here implies an active mechanism by the frogs to aquire bacteria which certainly is not the case, at least in general terms. Please rephrase.

Experimental design

no comment.

Validity of the findings

no comment

Additional comments

no fiurther comment

---

## Round 0.2 · Minor Revisions

Thank you for taking into account the reviewers previous comments. There are still some typos and sentences that need rewording. Please read through the manuscript carefully. I look forward to seeing and accepting the corrected manuscript

Reviewer 2 ·

Basic reporting

See below

Experimental design

See below

Validity of the findings

See below

Additional comments

While the authors have somewhat refined the manuscript following previous comments (both mine and other reviewers), I was disappointed to see that many aspects within the text had not been addressed; there are many examples of poorly worded sentences that present unclear statements to the reader. For example:

Line 22 - sentence unnecessarily long.
Line 25 - on not "in"
Line 26 - the effects of captivity, not "captive effects"
Line 27 - captive not "captivity"
Line 311 - needs a reference/references

Many comments within the document I returned after initial review, have not been addressed. For example:

Line 45 - The authors refer to "The amphibian conservation crisis", then list factors contributing to amphibian declines. Global amphibian declines require conservation effort to address these issues, but in the context of this sentence, the statement reads as though amphibian conservation is itself in crisis. An example of an amphibian conservation crisis would be perhaps a lack of funds, or manpower to facilitate action for a threatened species.

Line 63 - still unclear: has establishing ex-situ and disease-free colonies resulted in survival and reduced risk, or is this the aim of establishing ex situ and disease-free colonies?

Line 67 - why are ex-situ programmes almost the only option? This suggests there are alternatives.

While I fully accept that authors may, and are entirely entitled to choose to ignore or disregard reviewer comments, those pertaining to grammar, and clear understanding of the work being presented should be addressed.

I therefore suggest that the manuscript is re-submitted following further checks to improve grammar, remove ambiguity, and improve clarity of statements.

Reviewer 3 ·

Basic reporting

The authors have done a good job in revising the manuscript, and I found my main concerns all adequately addressed. I did not proofread the entire manuscript again line-by-line, but from glancing over the text I found only very few typos and sentences that would require improvement.

In the following, a few last corrections I would like to suggest:


271 --> classified in

326 --> remove semicolon

329 --> change belonged to belonging

323 --> ... did not reduce the diversity of bacteria as determined by culture-based assays ....

334 ---> rephrase to: "and these percentages were 55% ...." (or similar)

Experimental design

See Basic Reporting above

Validity of the findings

See Basic Reporting above

Additional comments

See Basic Reporting above

---

## Round 0.3 · accepted · Accept

Once the manuscript has been typeset into the PeerJ layout, please carefully check the wording of the paper as this will be the only occasion in which changes can be made.